# Optimisation of Energy Use in Bioethanol Production Using a Control Algorithm

**Jarosław Knaga \*, Stanisław Lis \***[ID]**, Sławomir Kurpaska, Piotr Łyszczarz and Marcin Tomasik**

Faculty of Production and Power Engineering, Agricultural University in Krakow, 30-149 Krakow, Poland;
slawomir.kurpaska@ur.krakow.pl (S.K.); piotr.lyszczarz@ur.krakow.pl (P.Ł.); marcin.tomasik@ur.krakow.pl (M.T.)
\* Correspondence: jaroslaw.knaga@ur.krakow.pl (J.K.); stanislaw.lis@urk.edu.pl (S.L.)

**Abstract:** In this work, the possibility of limiting energy consumption in the manufacturing process of bioethanol to obtain biofuel was analysed. For this purpose, a control algorithm has been optimised while retaining the good quality of the control signals. New in this study is the correlation of the control algorithm not only with the signal's quality, but also with the energy consumption in such an energy-intensive process as rectification. The rectification process in a periodic production system has been researched. The process was modelled on a test station with the distillation mixture capacity of 25 dm$^3$. For the optimization, the following control algorithms have been applied: relay, PID and PID after modification to I-PD. The simulation was carried out on a transfer function model of the plant that has been verified on a real object, a rectification column. The simulations of energy consumption and control signal's quality have been carried out in the Matlab®-Simulink environment after implementing the model of the research subject and control algorithms. In the simulation process, an interference signal with an amplitude of 3% and frequency of 2 mHz was used. The executed analyses of the control signal quality and the influence of the control algorithm on the energy consumption has shown some essential mutual relationships. The lowest energy consumption in the rectification process can be achieved using the I-PD controller—however, the signal quality deteriorates. The energy savings are slightly lower while using the PID controller, but the control signal quality improves significantly. From a practical point of view, in the considered problem the best control solution is the classic PID controller—the obtained energy effect was only slightly lower while retaining the good quality of the control signals.

**Keywords:** process control; bioethanol; energy use optimisation; computer modelling and simulation

## 1. Introduction

Due to the undoubted ecological benefits of the use of biofuels, including reduction of emissions into the atmosphere of harmful substances ($CO$, $CO_2$, $SO_2$), the creation of valuable by-products as a result of the production process of energy from biomass (such as rapeseed meal), the limitation of the occurrence of the phenomenon of "acid rain", and the relatively high biodegradability (in cases of disasters) of the raw materials and products used in the production process, the production of such fuels is an alternative to fossil fuels [1]. However, the decreasing attractiveness of transport biofuels generated with the use of materials derived from plant and animal sources (so-called first generation biofuels led to a search for solutions involving the production of 2nd and 3rd generation biofuels (so-called biofuels of the future) [2]. There are also many research projects being conducted which focus on improving the technologies used in the production of 1st generation biofuels. The use of ethanol as an additive to gasoline has many benefits for the natural environment. Experimental studies have shown that the use of such biofuels reduces emissions of $CO$, $SO_2$, $CO_2$, and toxic substances, while nevertheless leading to an increase in Nox emissions. These fuel mixtures contain oxygen in their composition, which increases the octane rating of the gasoline [3]. The lower energy value per unit of ethanol means that in comparison

with gasoline, usage is a dozen or more percent higher. An important aspect of the use of bioethanol, however, is the fact that the carbon dioxide which results from this process can be completely reabsorbed through processes of photosynthesis during the growth period of the subsequent batch of biomass intended to be used for the production of bioethanol. It is estimated that roughly 95% of global ethanol production is obtained through processes of the fermentation of raw materials.

Fornell and Berntsson [4] presented the manufacturing technology of bioethanol from lignocellulosic material. They observed that using the combined process of distillation and adsorption high-quality ethanol can be produced with a satisfactory energetic efficiency of the process.

To date, research papers on biofuels have focused on general economic issues in the production of bioethanol, the effects of using diverse raw materials in relation to intensification of the fermentation process, and optimal control of the machinery and devices used in the production of bioethanol. Additionally, in terms of economic research De Jong et al. [5] presented results of an analysis of production scale, type of transport used, integration of biomass supply and suppliers, as well as configuration of supply chains in order to reduce the costs of biomass production. To assess the feasibility of biofuel production, Grisolia et al. [6] introduced a bioeconomical index which takes into account both the course of the production process and the sustainable development of areas used for production of these fuels.

Regarding raw materials used and intensification of the fermentation process, Dharma et al. [7] defined the parameters of the process (taking into account the optimum process) of biodiesel produced from inedible vegetable oil, reaching the conclusion that the final product achieved meets quality norms. Baskar and Aiswarya [8] presented an overview of contemporary technologies regarding the use of raw materials and the selection of catalysers in the production of biodiesel, emphasising that the positive features of nano-catalysers make them an excellent choice for broad application. Nurfahmi et al. [9] analysed the potential of the leaves of the oil palm for use in the production of bioethanol (2nd generation biofuel) and analysed its quality; they determined that the variable feature was the different concentrations of stearic acid. Batog et al. [10] analysed the effects of using sorghum for the production of bioethanol; based on their research, they recommended a variety of this grain that was most effective due to its ethanol production capacity. Ciesielski and Grzywacz [11] presented a numerical cell growth simulation for the production of bioethanol, reaching the conclusion that the simulation model developed may be useful at the design and optimisation stages of control systems for the biomass fermentation process. Darvishi and Moghaddami [12] described a procedure for the intensification of the process (in both aerobic and anaerobic conditions) of generating bioethanol from beetroot molasses. Tgarguifa et al. [13] conducted studies and discovered the optimal parameters for a distillation column for the production of bioethanol with varying degrees of contamination of the substrate fed into the process. Da Costa and Normey-Rico [14] presented a procedure for the modelling, control, and optimisation of a fermentation column for the production of ethanol. They determined the decisive variables to be temperature, pH, and the speed at which the reactor is fed. As a result of their analysis, they proposed a solution for a control system intended to maximise production. Tripathi [15] presented the results of a study on the use of potato peelings for the production of bioethanol, reaching the conclusion that by correctly conducting preliminary processing, this type of biomass can be used as a substrate for the production of 2nd generation biofuels. Based on available research, Ganesan et al. [16], formulated an outlook for the use of algae (3rd generation biofuels) for the production of biofuels, reaching the conclusion that the main barrier to wider implementation of such solutions is process efficiency and productions costs. Zhao and Emptage [17] and Ozdingis and Kocar [18] have expressed the view that bioethanol (both 1st and 2nd generation) will allow for independence from supplies of fossil fuels and will meet environmental requirements for liquid fuels. Salim et al. [19] presented the results of research on the impact of the addition of thermophilic organisms to the starters

in the fermentation processes of lignocellulosic material in the production of bioethanol. Toora et al. [20] conducted a review of the literature in which they analysed the technologies necessary for the use of lignocellulose for the production of bioethanol, reaching the conclusion that further study is necessary on the process of the release of sugars and the intensification of the fermentation process. Aditiya et al. 2016] [21] conducted a review of the methods of production of 2nd generation bioethanol, paying special attention to bottlenecks in the process, including preliminary processing, hydrolysis, fermentation, and distillation. Mojovic et al. [22] assessed the current state and prospects for the development of technologies (based on currently available substrates) for the production of 2nd generation bioethanol; they reached the conclusion that by-products of the agricultural and food industries and selective cultivation of plants on land not used for agricultural purposes may significantly improve the financial results of production. Sarkar et al. [23] discussed available production technologies for 2nd generation bioethanol based on agricultural waste products.

To date, studies on the control of machinery and devices used in the production of bioethanol have concentrated on the selection of optimal process parameters. Regarding this, Ochoa et al. [24] presented the results of a study involving optimal process control in the production of biodiesel, reaching the conclusion that single- or double-layer control integrated in real time is from the economic point of view more favourable than when the process is controlled only with a conventional control loop. De Araujo et al. [25] developed a design for a simulation, optimisation, and control process structure for industrial conditions for the dosing of raw materials which takes into account the dynamics of heating of the material. Arifeen et al. [26] proposed an innovative configuration for a system for producing bioethanol using variable pressure distillation carried out in two separate columns, and as a result of the reduced heat losses between these columns they noted a considerable reduction of operating costs of the process. Pataro et al. [27] proposed an optimisation of the bioethanol production process involving the use of a control system providing dynamic optimisation which accounted for product quality and minimisation of energy losses, resulting in an improvement in these areas. Yadav and Verma [28] presented the results of a study involving modelling the operation of a multistage evaporator for the preliminary processing of biomass; with optimal regulation settings they achieved energy savings. De Freitas et al. [29] analysed the effects of the use of two systems controlling the feeding process for the bioreactors in bioethanol production. The authors stated that there was a significant correlation between the periodic feeding of the bioreactor and the productivity of the biomass and process duration.

Tgarguifa et al. [30] discussed the manufacturing technology of bioethanol on an industrial scale. On the basis of the performed researches (distillation process in combination with membrane separation—pervaporation) they noted that in this hybrid system the expenses can be lowered while obtaining a high-quality ethanol.

The brief review of the research literature presented above suggests that the issue of replacing fossil fuels currently being developed in a variety of academic centres. One of the criteria for the feasibility and profitability of the production of bioethanol is the ratio of energy input into the process to the amount of energy which can be obtained by the combustion of the bioethanol. The production of bioethanol is warranted in the case of raw materials from which the fuel produced has an energy value with a ratio of greater than one with regard to this. This ratio can be increased by using cheaper energy carriers in the production process or via optimal control of the supplied energy. The bioethanol production process can be divided into following phases: plant growing, production, dehydration, and mixing with gasoline and distribution. This implies that the highest production efficiency will be obtained when the expenses are reduced, including during dehydration. In the available literature there are no results originating from researches of a periodic process. From the analysis of the individual production phases, it could be concluded that the optimal rectification process control leads to the lowering of the energetic efficiency value in the manufacturing of bioethanol.

Optimal control of the rectification process in the production of bioethanol can lower the energy input required. Thus, it is reasonable to consider the analysis of the impact of the algorithm used to control the energy consumption of the process. A limitation was imposed in that the methodology used must allow for analysis by modelling and computer simulation which can be applied at the design stage of the technological system. Therefore, the aim of this study was to analyse energy usage and control quality when using a variety of controllers for the maintenance of optimal fermentation temperatures.

## 2. Materials and Methods

Realising the set objective, the following research hypothesis has been formulated: the control algorithm (the way the signal is formed by the controller) influences the energy consumption in the manufacturing process of bioethanol. The methodology used in the conduct of analysis of the possibilities for limiting energy usage in the process in question was based on the following procedure (Figure 1).

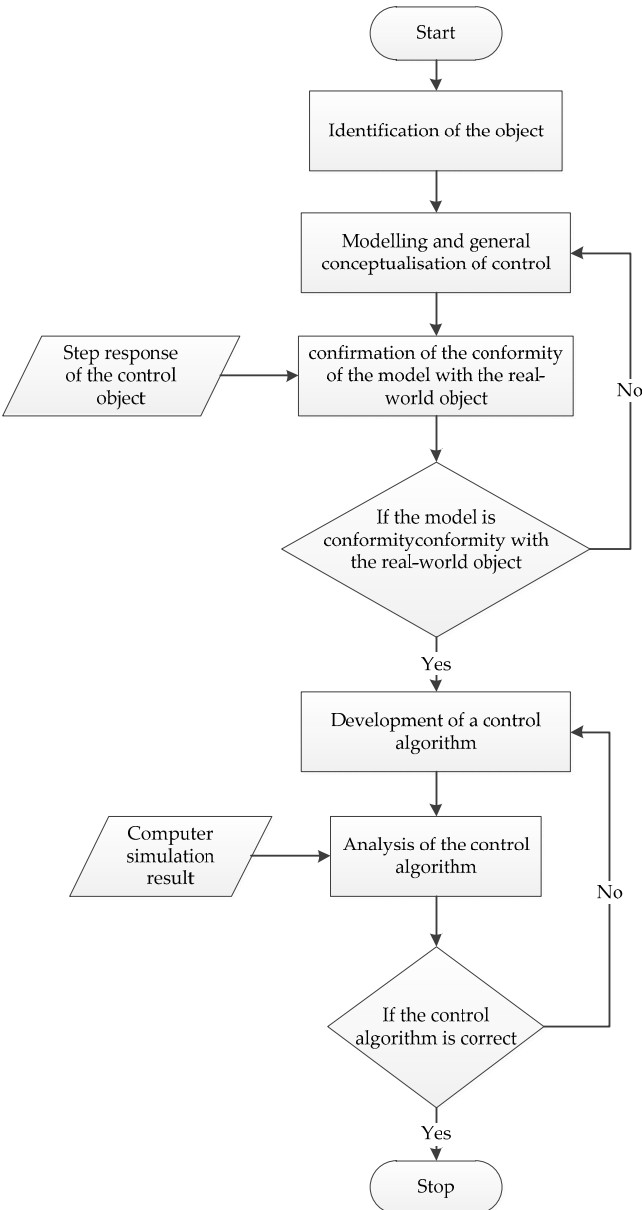

**Figure 1.** Iterative procedure for the optimisation of the control system as a function of limitation of energy usage in the process.

As part of the first stage of the procedure, an identification of the object to be controlled was performed via experimental determination of its dynamic characteristics. Next, a generic conception for control as a function of energy usage optimisation was formulated. Subsequently, a control algorithm was developed. At this stage, a simulation model of the object to be controlled was created. This was developed based on experimentally determined dynamic characteristics. The model was then fine-tuned by confirming the conformity of its behaviour with that of the object which it represented. The tuning algorithm involved the correction of model parameters during successive computer simulations in order to obtain the closest possible representation of the real object. Using the refined model of the object, a simulation model of the control system was created. This comprised the basis for computer simulation allowing for the selection of parameters of the control algorithm and analysis of the impact of the algorithm on energy usage in the process. In cases when an unsatisfactory result was obtained at this stage, it was possible to return to the conceptual stage, modify the conception, and repeat the analysis. If the algorithm is correct, the procedure ends [31–34].

The object to be controlled was a rectification column, a general outline of which is presented in Figure 2.

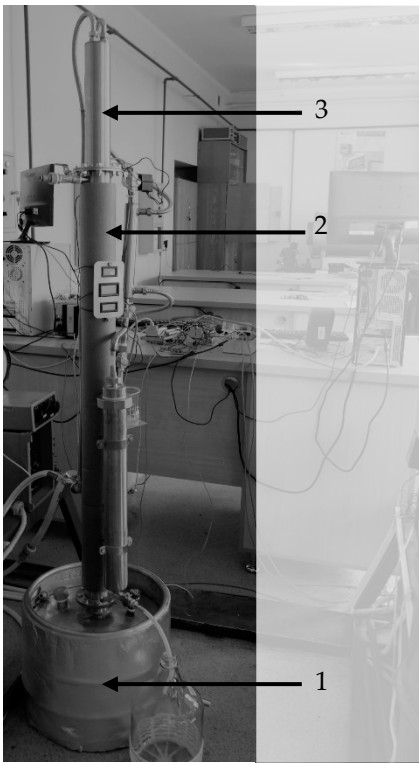

**Figure 2.** Rectification column: 1—reservoir with heating coil, 2—dephlegmator and 3—cooling head.

The device consists of a reservoir for the rectified liquid (1), with a dephlegmator located above this (2) and a cooling head at the top for the (3) liquid intended to be obtained. For the rectification process to take place, it is essential to provide heat, for which the heating coil located in the reservoir is responsible in this case (1). The basic operation of the column involves the use of counterflow contact of the distillation liquid flowing by gravitation down the dephlegmator with the rising vapours of the rectified mixture. Inside the dephlegmator, there are construction elements which increase the contact surface of the liquid with the vapours. During the process, about $\frac{1}{4}$ of the volume of the ethanol is directed for collection.

The preliminary stage of the development of the model of the control object was the experimental determination of the step response of the analysed object. To this end, a recording system was used.

During the experiment, the reservoir was filled with a distillation mixture with a volume of 25 dm$^3$. The mixture was in a steady state at a temperature of 20 °C. To determine the step response, a forced step increase in voltage was applied to the heating system with a total power output of $\Delta P$ = 4000 W. The reaction of the object to this forced increase in terms of change of temperature of the liquid $T$ was the step response. This was recorded until its course stabilised. In the experiment, a PT100 resistance sensor was used. Figure 3 illustrates the step response for the control object.

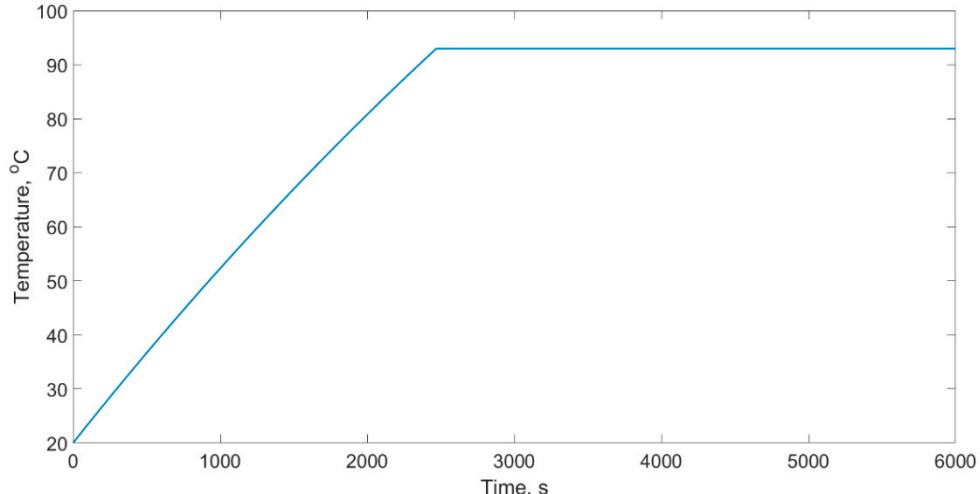

**Figure 3.** Step response of the control object.

This course (Figure 4) was the basis for the determination of the basic parameters of the system and the simulation model, as expressed in Equation (1),

$$(m \cdot c)_{zz} \frac{dT}{dt} = P_g - U_z \cdot A_z \cdot (T - T_{ot}) \tag{1}$$

where:

$(m \cdot c)_{zz}$—the reduced heat capacity of the reservoir calculated based on the measurement of the mass of the load and the reservoir (J·K$^{-1}$),

$U_Z \cdot A_Z = u_{zb}$—the reduced coefficient of heat loss of the reservoir calculated based on the materials used in its construction (12.03 W·K$^{-1}$),

$P_g$—the power of the heating coil (4000 W),

$T$—the temperature in the reservoir (°C),

$T_{ot}$—the ambient temperature (°C).

Equation (1) served to determine the established value and to verify the constant characteristics of the heat system comprising the studied object. Equation (1) only covers the process of heating without taking into account the process of evaporation or the thermal identification of the object. In order to simplify Equation (1), reduced quantities were applied vis-a-vis the heat capacity of the reservoir and the coefficient of loss, which at this stage allowed for the simplification of the modelling process without influencing quality.

For the steady state: $dT/dt$ = 0, Equation (1) can be expressed as (2)

$$0 = P_g - U_z \cdot A_z \cdot (T_{ust} - T_{ot}) \tag{2}$$

$$T_{ust} = \frac{P_g}{U_z \cdot A_z} - T_{ot} \tag{3}$$

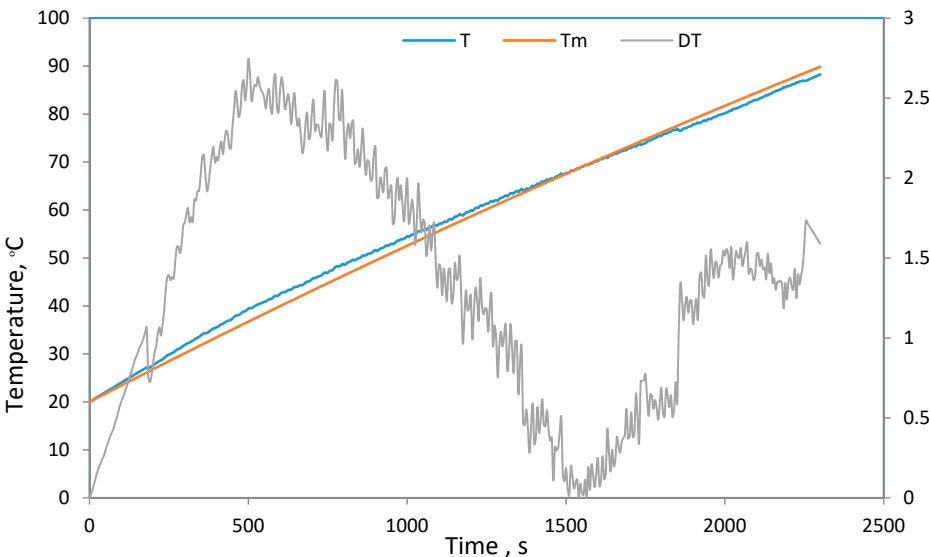

**Figure 4.** Verification of the temperature model with an indication of the course of absolute error.

Knowing the basic parameters of the reservoir at a constant power supply to the heating coil, the theoretically calculated temperature was established to $T_{ust}$ = 312 °C, which of course is a parameter characterising the reservoir while taking into account heat loss via thermal transmittance. Next, Equation (1) was modified by taking into account the constant power value $P_p$ intended for the process of evaporation after achieving the set temperature, which was expressed as in (4). Power $P_p$ was experimentally determined, and next the corrected value of the established temperature was determined using Equation (4) assuming a steady state of: $dT/dt = 0$:

$$(m \cdot c)_{zz} \frac{dT}{dt} = P_g - U_z \cdot A_z \cdot (T - T_{ot}) - P_p \tag{4}$$

The corrected value of the established temperature was $T_{k\_ust}$ = 275 at an average evaporation power input of $P_p$ = 445 W. Next, a model describing the change in temperature was empirically determined with the assumption that the object is a first order inertial system in line with the general Equation (5),

$$T = \frac{1}{a} \cdot \left(1 - e^{\frac{-a}{\beta} \cdot t}\right) \tag{5}$$

where:

$1/a$—the value of the established response,

$a/\beta$—the reciprocal of the time constant $1/T$.

Thus:

$$\frac{1}{a} = T_{k\_ust} \rightarrow a = 0.00364 \tag{6}$$

where $\beta$ = 29 was determined based on non-linear estimation, which for the analysed signal is presented in a chart (Figure 4). The largest absolute error did not exceed 2.75 °C, which is satisfactory and serves as a basis for developing the parameters of the transfer function.

For further analysis, a general form was assumed for the transmittance simulation model (7) $G(s)$ [35]:

$$G(s) = k_{ob} \frac{1}{Ts + 1} \tag{7}$$

The value of the time constant $T$ and gain were determined based on an analysis of a thermal model of the phenomenon under study as presented above. The static gain

coefficient of the object $k_{ob}$, calculated as the ratio of the change in temperature ($\Delta T$) to the change in power ($\Delta P$), is expressed in Equation (2).

$$k_{ob} = \frac{\Delta T}{\Delta P} = \frac{T_{k\_ust}}{P_g} = 0.0685 \tag{8}$$

The time constant was determined based on the developed Equation (7);

$$T = \frac{\beta}{a} = 7970 \tag{9}$$

Thus, the final form of the model of the object of study was (10).

$$G(s) = 0.0685 \frac{1}{7970s + 1} \tag{10}$$

where: *s*—the Laplace operator.

Before beginning the analysis of the impact of the control algorithm on energy usage, the quality of the control of the modelled system was determined (Figure 5). As the criteria for assessment, the integral quality indicators *WJS*1 and *WJS*2 were used, where *WJS*1 is the integral of the value of the absolute error of regulation (11), while *WJS*2 is the integral of the value of the absolute derivative of the control signal (12) [36,37].

$$WJS1 = \int_{t_p}^{t_f} |e| dt \tag{11}$$

$$WJS2 = \int_{t_p}^{t_f} \left| \frac{du}{dt} \right| dt \tag{12}$$

where: *e*—the regulation error, $\frac{du}{dt}$—the derivative of the control signal, *t*—time, $t_p$— beginning of the control time interval, $t_f$—end of the control time interval.

*WJS*2 provides information on the dynamics of the control signal, while the value of the *WJS*1 indicator provides information on the quality of control; the lower this value is, the better is the control quality.

In line with our assumptions regarding this development, an attempt was made to select a control algorithm as a function of lowering energy usage in the controlled process. By algorithm, we understand here the method of shaping the control signal. The controller is responsible for this operation in the control system. Thus, by changing the type of controller, changes in the control algorithm are obtained. The adopted methodology assumes an analysis of the system at the prototype stage via a computer simulation using the developed model of the object and the control system. It is assumed that based on the results of the computer simulation, it will be possible to determine the control quality and assess the impact of the control algorithm on energy usage in the rectification process. Control using three types of controllers: relay, PID and I-PD, was subjected to analysis. Their algorithms were implemented in the Matlab-Simulink programme as block schemes as illustrated in Figure 5.

The symbols presented in the Figure refer to the following: In—input; Out—output; *kp*—proportional gain; *Ti*—integral time (doubling); *Td*—differential time (advance); *β*— gain in the I-PD controller's proportional band; *r*—value of the input temperature; *e*— regulation error; *u*—control signal; and *y*—controlled value.

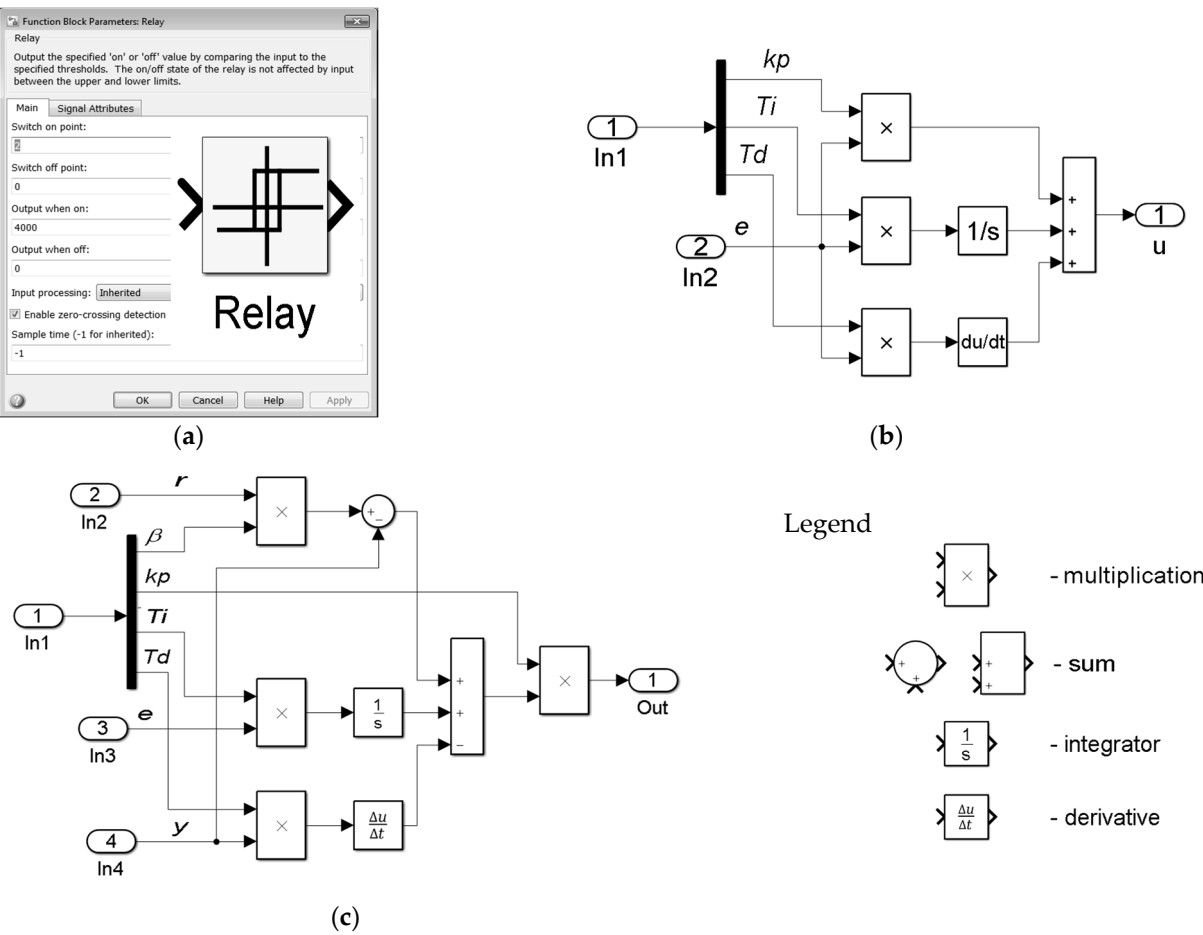

**Figure 5.** Controller block scheme: (**a**) relay, (**b**) PID and (**c**) I-PD.

Relay is the simplest type of analysed controller (Figure 5a). In this type of controller, the control signal has only two values, minimum and maximum. This type is recommended for use with thermal objects. The shortcoming of the classic PID controller (Figure 5b) is its gain in noise, which results in a deterioration of regulation quality and an increase in the probability of the occurrence of oscillation in the output signal [38]. The characteristic feature of the I-PD controller (Figure 5c) is the presence at the input stage of differential signalling of the controlled value $y$, and in the proportional gain band of the measured value $\beta$, which fits into the interval <0, 1>. This defines the type of signal supplied to the input of the proportional element.

Based on the model of the object (Equation (10)), a simulation control model was created. This comprised the basis for a computer simulation allowing for the selection of a control algorithm and optimisation of this algorithm in terms of reducing energy usage in the process. In the Matlab-Simulink environment, three versions of the control system were implemented, configured with different types of controllers: relay, PID and I-PD (Figure 6) [39,40].

The Setpoint represents the set temperature, Controller is the transmittance of the controller, and transfer fcn and transport delay represent the control object. In the scheme, in the control loop there is also an additional block which is not part of the structure of the system—this is the signal generator. This block simulates noise effects. Its presence during simulation studies allows for the analysis of the impact of the noise signal on control quality. The remaining symbols are discussed in the description for Figure 5.

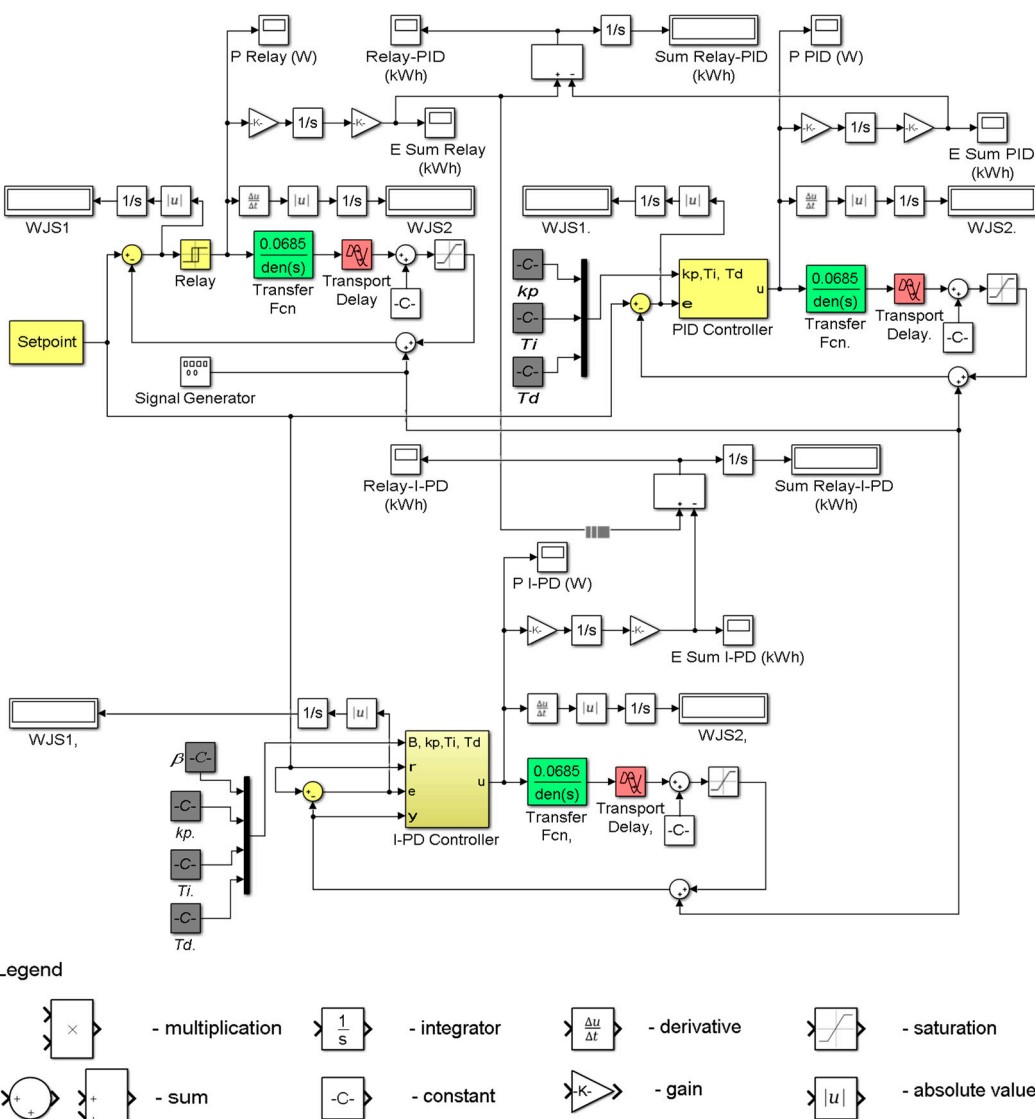

**Figure 6.** Block schemes of the integrated simulation models of the control systems with relay, PID and I-PD controllers.

## 3. Results

The key aim of the study conducted was to determine the impact of the control algorithm on energy usage in the process in question via computer simulation using a model of the control system. For this purpose, the course of the control signal at the controller output was analysed. This signal, when combined with the power of the actuator (4 kW) allows the energy usage to be estimated. The values which are obtained in this way are of an estimated nature and allow for a preliminary selection of the type of controller as a function of reducing energy usage.

For the simulation, a process duration of 2.5 h was assumed, which is equivalent to real conditions in the study conducted. The set signal was chosen according to an algorithm predicting the increase and maintenance of the temperature of the liquid in the reservoir of the column (Figure 2) at a level ensuring the correct course of the process.

Initially, the operation of the system was analysed (Figure 6) without noise. The results of the simulation are illustrated in Figure 7.

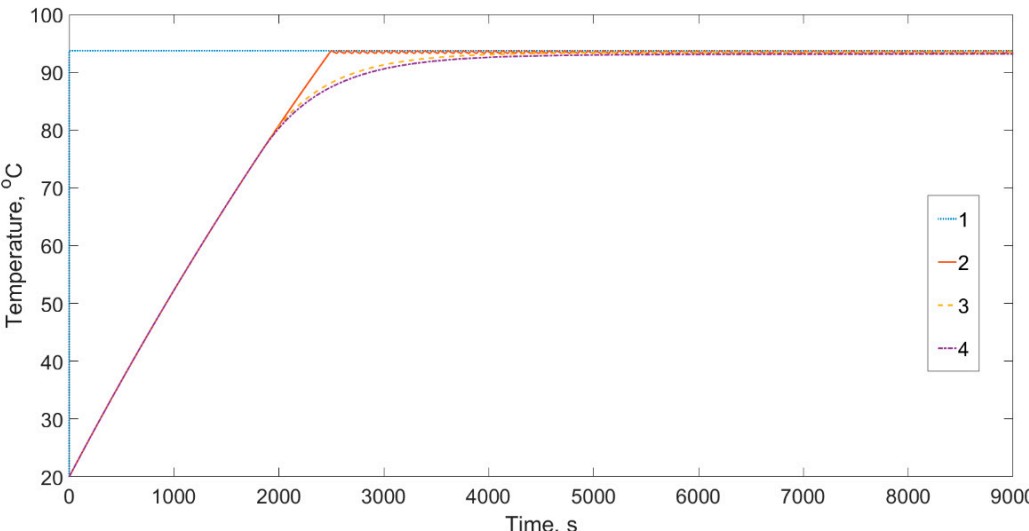

**Figure 7.** Results of the simulation of controlling the course of temperature—no noise: 1—value of the set temperature, 2—output signal of the control system with a relay controller, 3—output signal of the control system with a PID controller and 4—output signal of the control system with an I-PD controller.

To highlight the character of the course illustrated in the chart (Figure 7), a part of the process was isolated and presented in Figure 8. In the selected time interval, the regulated value is in a stabilisation phase.

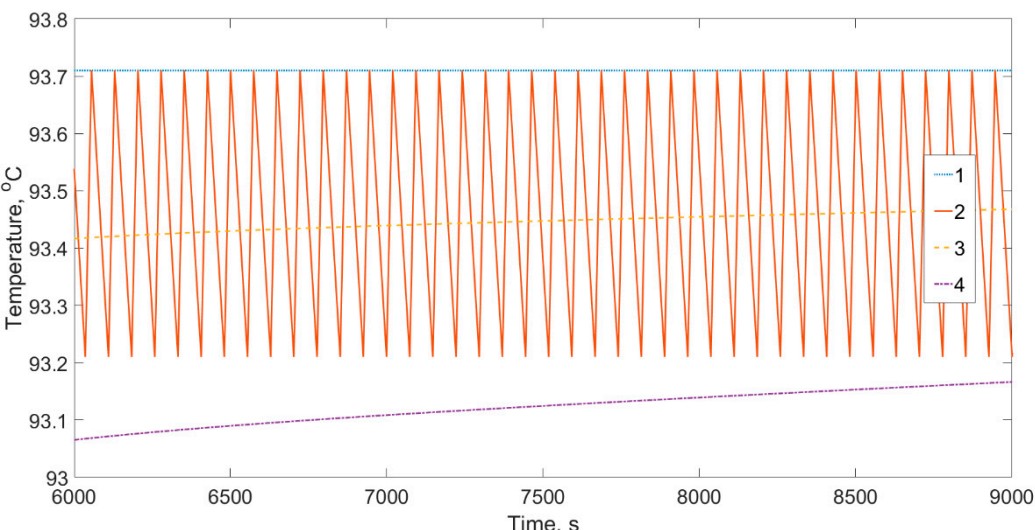

**Figure 8.** Results of the control simulation for the isolated time interval—no noise: 1—value of the set temperature, 2—output signal of the control system with a relay controller, 3—output signal of the control system with a PID controller and 4—output signal of the control system with an I-PD controller.

The integral values of the quality indicators for the controllers studied are presented in Table 1.

**Table 1.** Integral values of the control quality indicators in optimal conditions.

| Indicator / Controller | Relay | PID | I-PD |
|---|---|---|---|
| *WJS*1 | $8.89 \times 10^4$ | $9.33 \times 10^4$ | $9.63 \times 10^4$ |
| *WJS*2 | $1.30 \times 10^7$ | $4.70 \times 10^4$ | $4.56 \times 10^4$ |

An analysis of the courses of signals 2–4, and the values of the indicators *WSJ*1 and *WSJ*2 (Table 1) presented in Figures 7 and 8 suggests that the algorithms of the studied controllers during the computer simulation study for ideal conditions (without noise) ensure similar control quality. These courses do not differ significantly from the signal of the set value represented by curve 1. Th temperature in the reservoir is maintained correctly. An analysis of the chart in Figure 8 allows for the observation that the signal from the system with the relay controller is peculiar to the applied algorithm, while the curve representing the control system with the I-PD controller marked with the number 4 differs slightly from that of the system with the PID controller. In this case, the temperature is lower than the set value (0.3 °C). This however does not result in a significant deterioration of control quality (indicator *WSJ*1 in Table 1). Based on the integral values of the *WSJ*1 indicators, it can be stated that the best control quality is provided by the relay controller, as a result of the high inertness of this type of process. The PID controller delivers a somewhat worse result, while the I-PD controller differs significantly from the PID algorithm. Additionally, it should be emphasised that a better result for the I-PD controller was obtained with a load value of $\beta = 1$, which is directly related to the algorithm of this controller. A change in the value of $\beta$ towards 0 resulted in a deterioration of the control quality.

Subsequently, the control process was analysed taking into account a noise signal with a random course at a frequency of 2 mHz and an amplitude of 3% of the maximum value of the set signal. A fragment of the course of the noise signal is illustrated in Figure 9.

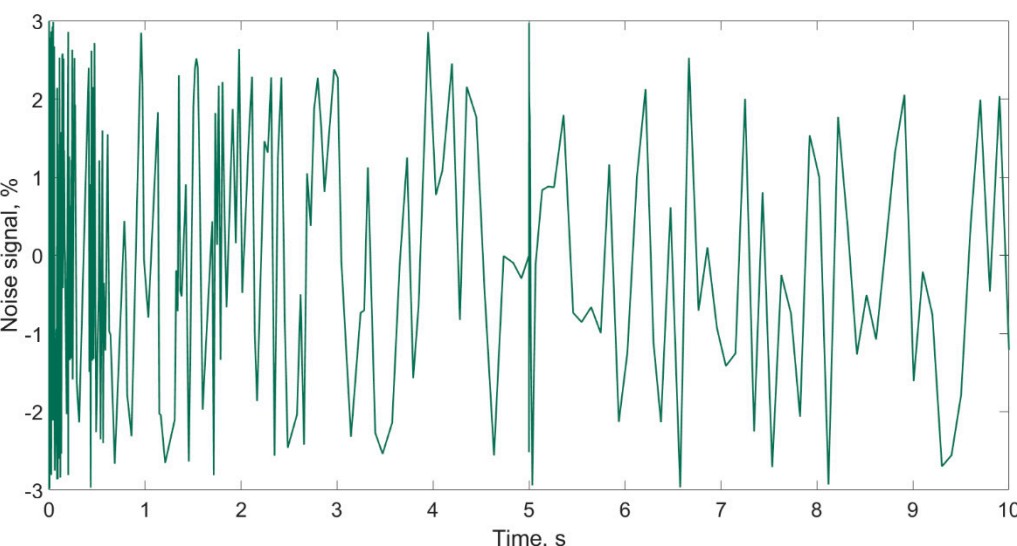

**Figure 9.** Noise signal with a random course at a frequency of 2 mHz and an amplitude of 3%.

The results of the simulation for the modelled system subjected to the effects of a noise signal (Figure 9) are presented in Figure 10, where the conditions of real simulation of noise in the course of the temperature there is a noticeable effect of over-regulation of the temperature for the relay controller (the red line exceeds the set value).

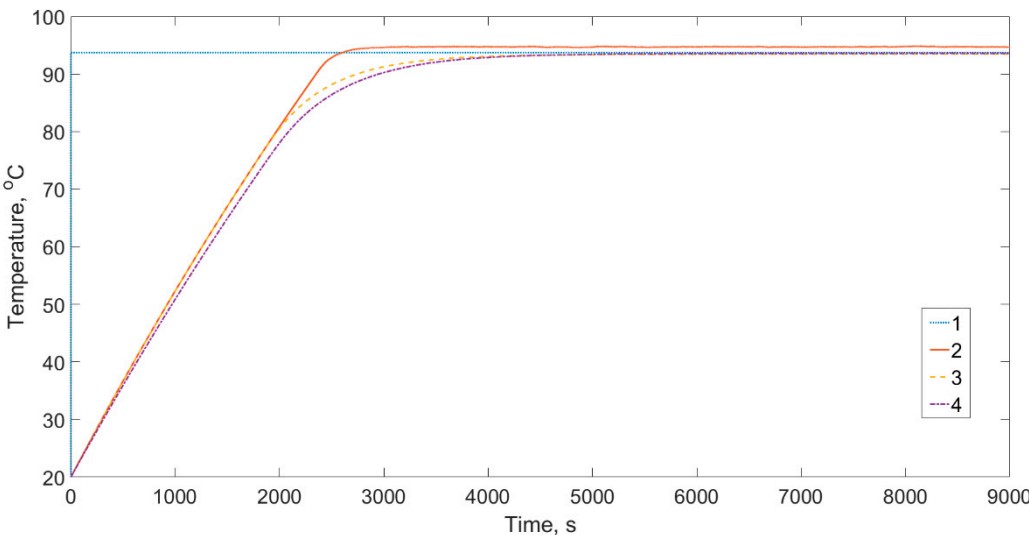

**Figure 10.** Results of the simulation of control with a random noise signal with a maximum amplitude of 3%: 1—set value of the temperature 2—output signal for a control system with a relay controller, 3—output signal for a control system with a PID controller and 4—output signal for a control system with an I-PD controller.

For the simulation in question, values of the indicators *WSJ*1 and *WSJ*2 were determined, as presented in Table 2.

**Table 2.** Integral values of the indicators of control quality for a noise signal with a random course and maximum amplitude of 3%.

| Indicator / Controller | Relay | PID | I-PD |
|---|---|---|---|
| *WJS*1 | $9.83 \times 10^4$ | $9.99 \times 10^4$ | $1.05 \times 10^5$ |
| *WJS*2 | $1.09 \times 10^7$ | $1.78 \times 10^7$ | $3.85 \times 10^8$ |

An analysis of the results of the simulation for the process with a noise signal (Figure 9), illustrated in the chart in Figure 10 and described with integral indicators (Table 2), allows for the observation that there is a logical correlation between the noise signal and the integral values of the quality indicators *WJS*1 and *WJS*2. Namely, for the controller algorithms studied, the noise signal results in an increase in the value of these signals, quite understandably; however, their differences with regard to the relay controller are significantly smaller in comparison to the test without noise. The highest *WSJ*1 indicator value was obtained for the I-PD controller, resulting from the fact that the course of the temperature for this controller (curve 4 in Figure 10) slightly differs from the set value to a greater extent than in the case of the other controllers. The values of the *WSJ*1 indicator for the relay and PID controllers are similar, indicating a similar control quality in the presence of noise. In the case of the system using the relay controller algorithm, we observe the lowest dynamics of the control signal (information provided by the value of the *WJS*2 indicator) when compared to the other controllers.

After analysing control quality, the key part of the simulation studies was conducted, aiming to determine the impact of the control algorithm on energy usage in the rectification process. The analysed courses of temperature control signals via the power of the actuators (without noise in the system) are presented in Figure 11.

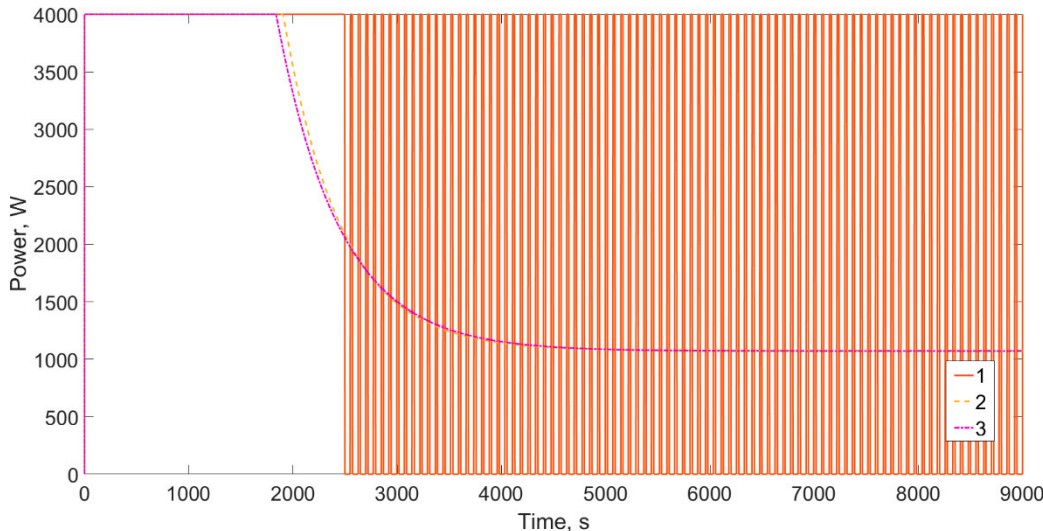

**Figure 11.** Course of the control signals from the different controllers (no noise): 1—relay, 2—PID and 3—I-PD.

To highlight the character of the courses illustrated in the chart (Figure 11), a part of the process was isolated and presented in Figure 12. In the selected time interval, the regulated value is in an unstabilised phase.

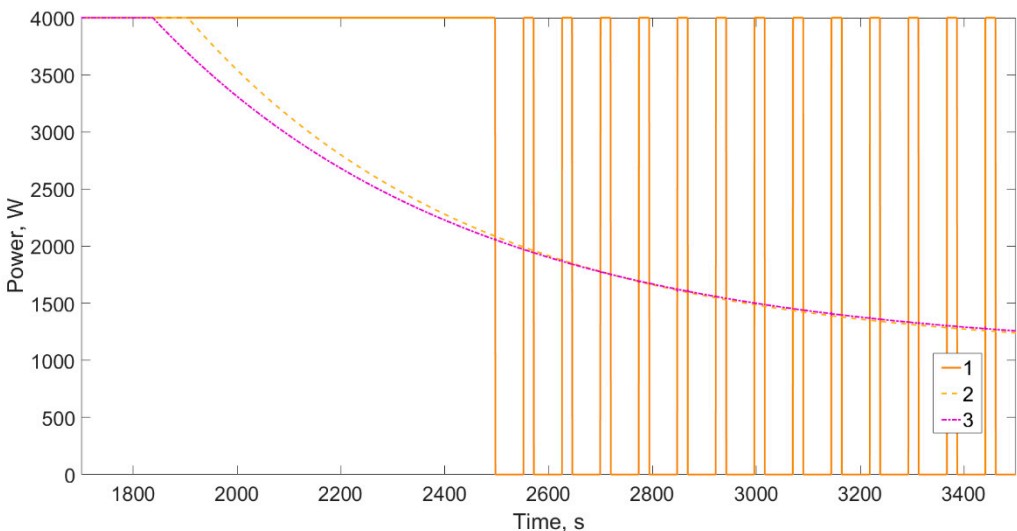

**Figure 12.** Course of the control signals from the different controllers for the isolated time interval (no noise): 1—relay, 2—PID and 3—I-PD.

An analysis of the powers of the heating element indicates that these reflect the control signals of the applied controller algorithms. Up to a 33 min time interval, the power for all the controllers is maintained at the level of 4 kW. After this time, the signal of the relay controller, in order to maintain the set temperature, alternates between two values, 0 kW and 4 kW. Conversely, the values of the signals for the PID and I-PD controllers in the second part of the analysed time interval are significantly reduced to a power of 1.1 kW. The total energy usage for the controllers studied is 4.75 kWh, and their courses do not differ significantly in the absence of noise (Figure 13).

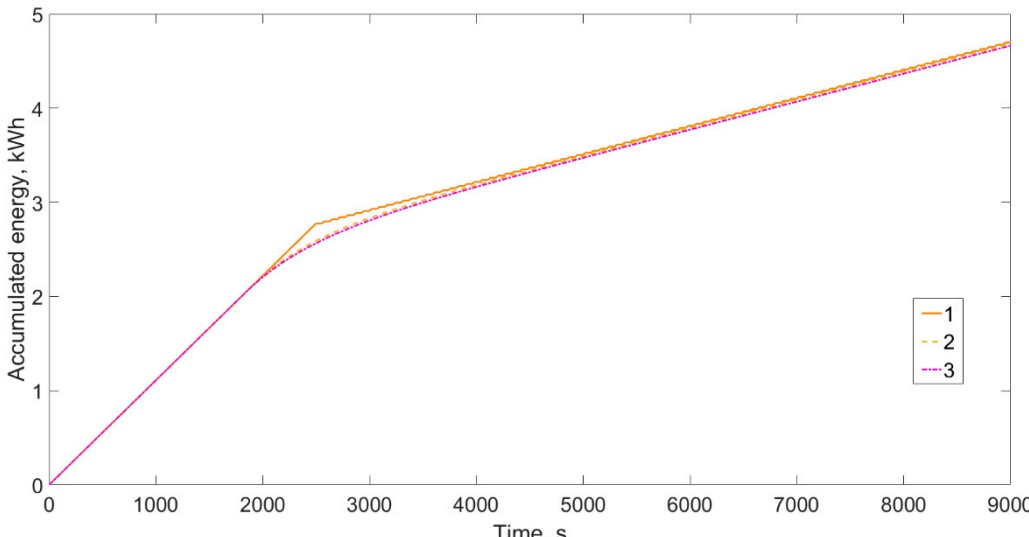

**Figure 13.** Course of cumulative energy usage for control systems with the controllers (no noise): 1—relay and 2—PID, 3—I-PD.

Just as in the case of the simulation for ideal conditions, the impact of the control algorithm on energy usage was analysed for the effect of the presence of a noise signal (Figure 9). The results of the simulation indicate a high level of dynamics for the control signal caused by the effect of noise.

Total energy usage for the analysed controllers during the simulation of the operation of the system with noise is presented in the chart (Figure 14).

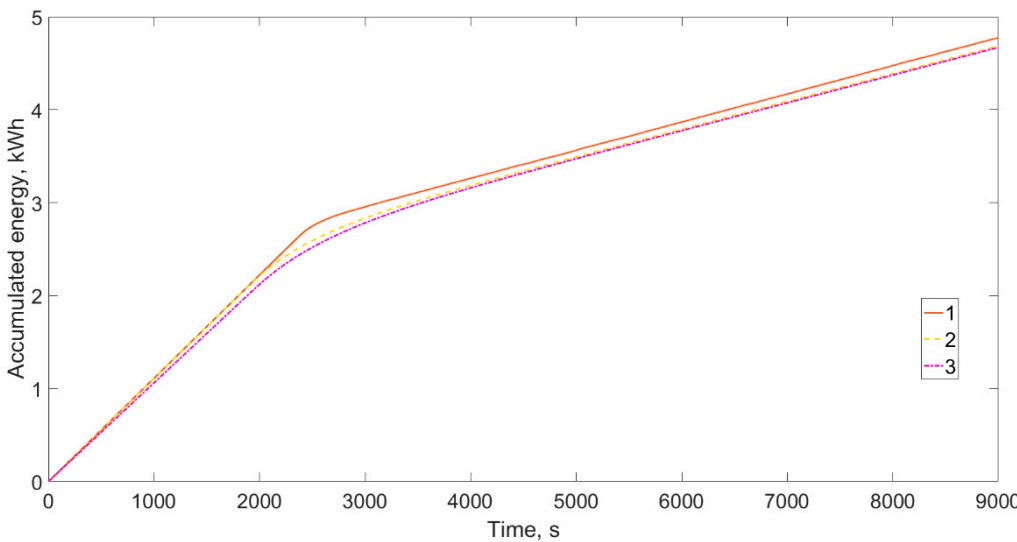

**Figure 14.** Total energy usage for control systems in the presence of a random noise signal with a maximum amplitude of 3% for the different controllers: 1—relay and 2—PID, 3—I-PD.

Based on the chart presented in Figure 15, it can be stated that the energy usage for particular control algorithms differs, and is as follows for the different controllers; relay—4.81 kW, PID—4.71 kW, I-PD—4.7 kW.

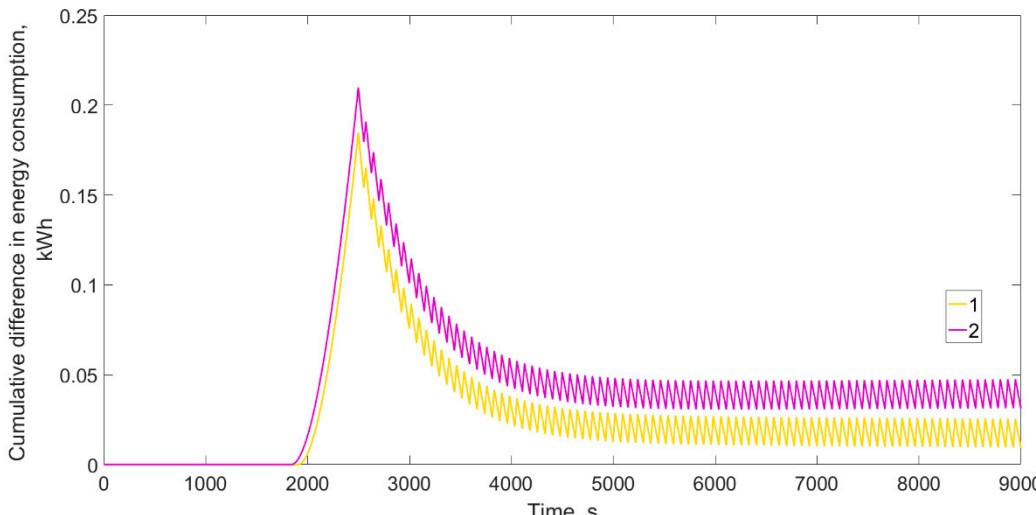

**Figure 15.** Course of the cumulative energy usage difference for control systems without noise 1—(relay—PID) and 2—(relay—I-PD).

To assess the energy efficiency of the analysed control systems, in the next part of this paper use is made of the difference in cumulative energy usage, and this difference is determined with respect to the energy usage for the relay controller. Thus, in further analyses the terms relay-PID and relay-I-PD indicate the difference in cumulative energy usage between the applied relay and PID controllers, and between the relay and I-PD controllers. The results of these analyses are presented in absolute and relative systems, as shown in the charts (Figures 15 and 16).

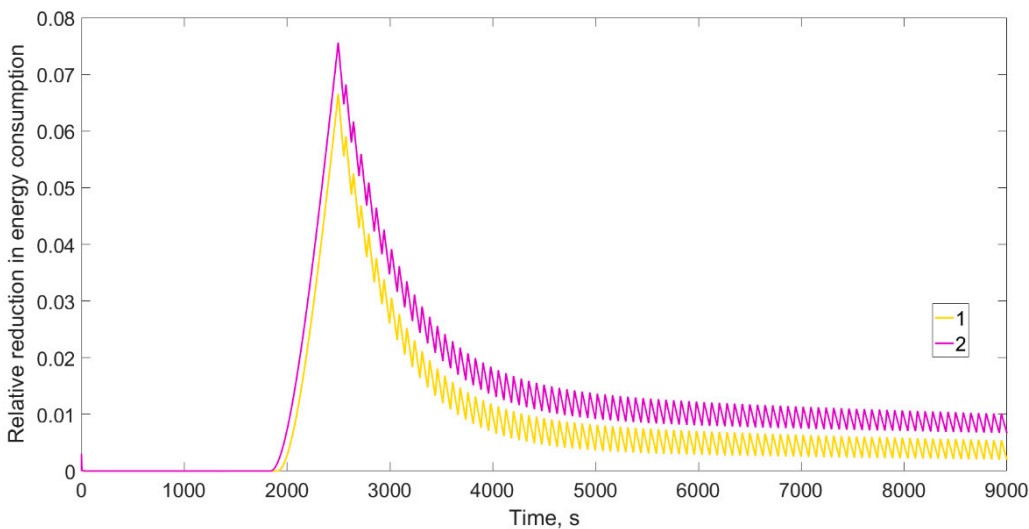

**Figure 16.** Course of the relative decrease in energy usage for control systems without noise 1—(relay–PID) and 2—(relay—I-PD).

The greatest differences in energy usage appeared when shifting from the state of preheating the load to maintaining a constant temperature, and these were observable even without noise, as is illustrated in the charts (Figures 15 and 16) with a clear peak. However, the values of the reduction in energy usage in the system do not exceed 0.8% for PID and 0.4% for I-PD and do not thus significantly differentiate the solutions studied in terms of energy usage.

A similar analysis was conducted for the controllers in question with the introduction of noise in the feedback loop, as is illustrated in the charts (Figures 17 and 18).

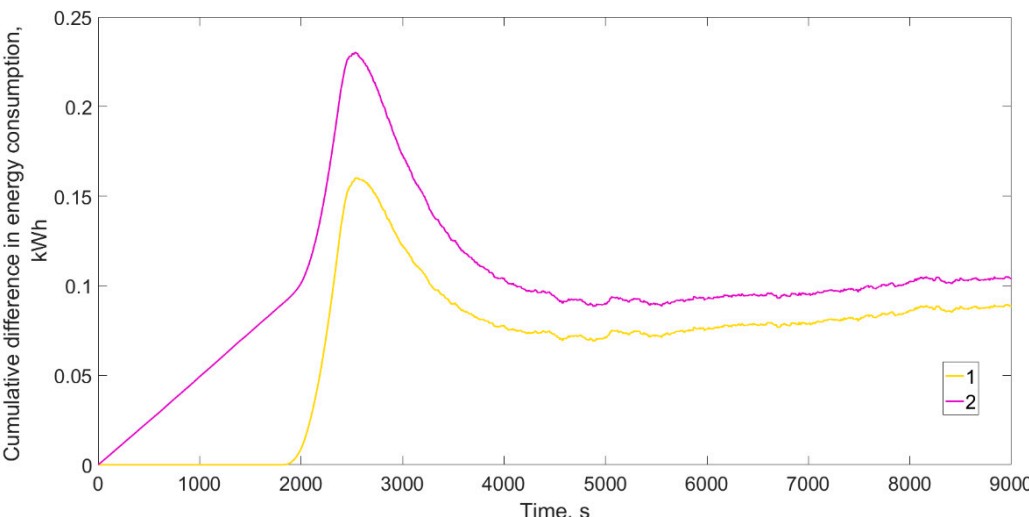

**Figure 17.** Course of the cumulative energy usage difference for control systems with a noise of 3%: 1—(relay—PID) and 2—(relay—I-PD).

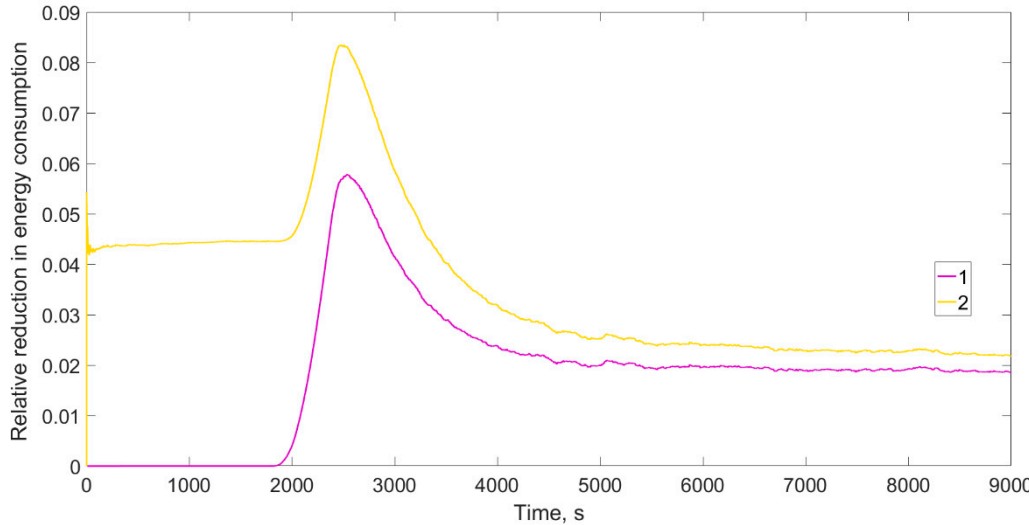

**Figure 18.** Course of the relative decrease in energy usage for control systems with a noise of 3%: 1—(relay—PID) and 2—(relay—I-PD).

As with forced noise, the greatest differences in energy usage appeared when shifting from the state of preheating the load to maintaining a constant temperature, as is illustrated in the charts (Figures 15 and 16) with a clear peak. However, it is important to note that in the case of the PID controller, the characteristics of heating the load in terms of energy usage do not differ from that of the relay controller, in contrast to the I-PD controller, where we also observe a slight decrease in energy usage of roughly 4% in the initial phase. Regardless of whether we refer to the PID or I-PD controller, in the steady state of the temperatures of the rectification process the energy usage slightly decreases along with the duration of the process. On the other hand, the relative decrease of the energy usage in the rectification process remains at an unchanged level, respectively 1.9% for the PID controller and 2.1% for the I-PD controller. In terms of energy usage, the better solution to be used in

the rectification process is the I-PD controller, while in terms of control signal quality the PID is superior, as with a noise of 3% in the control signal its quality is comparable to that of the relay controller.

The presented analysis of control signal quality at different imposed interference in the amplitude range of 5% and frequency depending on the process being the control subject is a frequent practice [36,37,41] that shall indicate the optimal solution for the process control. Analysing the heating process control of a foundry furnace, the authors observed that the fuzzy control is the best solution, also regarding the use of the power. Analysing the results of these studies, it can be noticed that the comparison of the obtained indicators *WJS*1 and *WJS*2 is done only within the analysed process and its control. From the overview of the available research results it follows that currently there are no reports about the control of the periodic rectification process in micro installations, especially in conjunction with energetic expenses. In the considered process, classic control systems and the modified I-PD controller have been analysed. Regarding the controlling quality, the best effects, independently of the interference, have been obtained for the simplest solution "relay". However, from the energetic point of view, the modified I-PD controller provided the best results, simultaneously being the worst solution regarding the control quality. From the practical point of view, the best solution from those analysed is the classic PID controller that generated a slightly lower energetic effect while retaining a good quality of the control signals. Thus, it can be concluded that the presented methodology of the two-criterial optimization of the control system, implemented on the example of the bioethanol rectification process, shows its high suitability for energy absorbing processes, such as bioethanol production. Depending on the raw materials of which the bioethanol is manufactured, for example, grains and corn, rectification constitutes less than 20% of the energetic expenses for the production [42]. In the case of molasses, the highest energy expenses for the rectification process reach 80%. The reduction of the energy consumption by 2% in the rectification process can bring measurable benefits for the environment, using the optimal control system.

## 4. Conclusions

The assessment of the control system in a two-criterial system, i.e., energy consumption and quality of control system in the periodic rectification process, on the example of small installation for periodic production of bioethanol, clearly indicated that the energy consumption can be reduced for more advanced control algorithms. However, it decreases the quality of the quality of the signals controlling this process in case of interferences or weak quality of the applied temperature sensors.

Based on the defined assumptions, after conducting a simulation process for the control algorithms relay, PID and I-PD, it was determined that:

1. The best control algorithm in terms of signal quality is the relay, regardless of whether the analysis was conducted without noise or with a noise of 3% and a frequency of 2 mHz entered into the feedback loop. In each of the analysed cases, the indicators *WJS*1 and *WJS*2 reached their lowest values.
2. The best solution for control in the process studied in terms of energy usage was the I-PD algorithm, which allowed for a reduction of energy usage of 0.8% for the simulation without noise and 2.1% for the simulation with a noise of 3% at a frequency of 2 mHz.
3. The worst solution in terms of signal quality was the I-PD algorithm regardless of the *WJS*1 and *WJS*2 indicators assumed, which reached their highest values.
4. The PID regulation algorithm applied to the periodic rectification process allowed for the energy usage to be reduced by 1.9%, which is slightly worse that the I-PD algorithm in terms of energy usage. Nevertheless, the quality of the control signal for the indicators studied did not significantly differ from the control signal quality for the relay controller.

The development of the control algorithms, especially under consideration of the lower and lower costs of their implementation, should stimulate research on more complex criteria of their application, where energetic expenses should be an essential factor. Activities leading to the reduction in energy consumption, especially in energy-intensive processes, are prioritized for the natural environment protection.

**Author Contributions:** Conceptualization, S.L., J.K. and P.Ł.; formal analysis, S.L. and J.K.; methodology, J.K., S.L., S.K., M.T. and P.Ł.; software, S.L. and P.Ł; validation, P.Ł., J.K. and S.L.; writing—draft preparation, S.L., J.K., S.K. and M.T.; writing and editing, S.K., S.L. and J.K. All authors have read and agreed to the published version of the manuscript.

**Funding:** The publication was financed with a grant from the Ministry for Higher Education for statutory activities.

**Institutional Review Board Statement:** Not applicable.

**Informed Consent Statement:** Not applicable.

**Data Availability Statement:** Data is contained within the article.

**Conflicts of Interest:** The authors declare no conflict of interest.

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
