# Peer review of "Optimisation of Energy Use in Bioethanol Production Using a Control Algorithm"

_processes, doi:10.3390/pr9020282_

Round 1
Reviewer 1 Report
The proficient usage of English is required for a journal paper. There are many 'too-long' sentences in your work. E.g., the first sentence in the abstract, four sub-sentences are included: this study analyses a possibility; limit energy used in the production process of bioethanol; limit energy by optimising the control algorithm; maintain quality of control signals. I feel hard to achieve these meanings. Try to use concise expressions.
There are also some confusing word use in the context. E.g., in line 149, why use 'ration' instead of 'ratio'. In line 214, 'Tot' shoud be 'Tot'.
The figures should also be improved: In Fig.1, why not use a classical flowchart, some arrows does not point to an element, and some elements are not connected with arrows. Result figures require clear legends rather than single characters or numbers.
Explanations are required for some figures, especially for figure 5 and 6. They are the designs of relay, PID and I-PD, core of the control algorithm. For readers with no background knowledge of Simulink, there should be better presentations. The figures in the result section look similar so an explanation of their meanings, or a comparative analysis should be performed.
Even the introduction section is long, background and motivation of the work should be pointed out clearly.
Author Response
Response to Reviewer
Thank you for the valuable comments that undoubtedly emphasise the scientific character of this work and let better understand the problem we wanted to explain. We hope the corrections we made will be satisfying.
Point 1: The proficient usage of English is required for a journal paper. There are many 'too-long' sentences in your work. E.g., the first sentence in the abstract, four sub-sentences are included: this study analyses a possibility; limit energy used in the production process of bioethanol; limit energy by optimising the control algorithm; maintain quality of control signals. I feel hard to achieve these meanings. Try to use concise expressions.
Response 1: According to the suggestion, the Abstract and the Introduction have been redrafted.
Point 2: There are also some confusing word use in the context. E.g., in line 149, why use 'ration' instead of 'ratio'. In line 214, 'Tot' shoud be 'Tot'.
Response 2: According to the reviewer’s comments, editorial improvements needed due to Authors’ oversight have been done.
Point 3: The figures should also be improved: In Fig.1, why not use a classical flowchart, some arrows does not point to an element, and some elements are not connected with arrows. Result figures require clear legends rather than single characters or numbers.
Response 3: According to the suggestion, appropriate corrections have been made in the paper.
Point 4: Explanations are required for some figures, especially for figure 5 and 6. They are the designs of relay, PID and I-PD, core of the control algorithm. For readers with no background knowledge of Simulink, there should be better presentations. The figures in the result section look similar so an explanation of their meanings, or a comparative analysis should be performed.
Response 4: In the corrected paper, in order to explain the function of the basic blocks visible in the Figures 5 and 6, a legend has been added to the figures.
Point 5: Even the introduction section is long, background and motivation of the work should be pointed out clearly.
Response 5: The corrected paper has been supplemented with following statements:
One of the possibilities for the limitation of energy consumption in the manufacturing process of bioethanol is the optimization of the control algorithm. In the scope of the control, especially the control quality, extensive literature is available. Hereby not only classic control algorithms are considered, but also modifications of them and the artificial intelligence. The literature is also comprehensive regarding the setting optimisation of these controllers and their influence on the control signals, which is a one-criterial optimization of the control algorithms. New in this paper is the introduction of the second criterion for the assessment of the subject algorithms, corresponding to the energy consumption in energy-intensive processes. Besides, in a sense, new is the selection of the researched object, i.e. a periodically working rectification column that can be used e.g. by farmers to produce biofuel from waste. The Authors believe, researches in this area have not been carried out. Thus, it should be considered as reasonable to carry out an analysis of the influence of the applied control algorithm, among others, on the energy intensity of the process. Hereby a condition has been imposed that the methodology has to enable an analysis by modelling and computer simulation at the stage pf the technical system designing. New in the presented study is the analysis of the energy consumption reduction by selecting the optimal control algorithm. Besides, new is the execution of the analysis in question at the designing stage by modelling and computer simulation.
Reviewer 2 Report
In this paper, the authors presented three controllers (relay, PID and I-PD) performance on a lab-scale rectification process with the intention of reducing the energy consumption of bioethanol processes. To implement these controllers in Simulink, a transfer function was identified from the experimental tests to represent the system. This paper’s language and flow need to further improve for better readability. Overall, this paper lacks novelty from control algorithm and application complexity aspects. It would be more interesting to have some more advance control algorithm or tuning algorithm since there are already extensive theories and guidelines on how to implement and tune a PID-level controller. Thus, this paper is suggested to be reconsider after major revision with the following comments.
- Introduction: a complete literature review should be conducted to focus on the “bioethanol process” rather than a more wider area of ”biofuel processes”. I think that authors interchanged the term “biofuel” or “biodiesel” with “bioethanol”. Also introduction part should present the knowledge gaps of process control for bioethanol processes and emphasize the novelty of this work.
- Is rectification column the biggest energy consumer in the bioethanol processes? What is the typical fraction of the energy consumption in a regular bioethanol plant? I think rectification column is a very common process unit in most of the chemical processes and it is not hard to control rectification column. Please justify why chose rectification column as a control application.
- The implemented controller in this work is to drive the bottom temperature to setpoint. It is much more meaningful to produce on-spec bioethanol (e.g., ethanol concentration >95%). Have the authors tried to deign the controller for a more flexible operation of rectification column.
- How did you tune control parameters (e.g., kp, Ti, Td) for each implemented controller?
- Energy usage in the result part is the average energy usage during the whole time horizon or the energy usage at the reached steady state.
- The transfer function representing rectification column is developed from lab-scale experimental tests. Please comment that how close/different this model is from other model in the literature.
Author Response
Response to Reviewer
Thank you for the valuable comments that undoubtedly emphasise the scientific character of this work and let better understand the problem we wanted to explain. We hope the corrections we made will be satisfying.
Point 1: Introduction: a complete literature review should be conducted to focus on the “bioethanol process” rather than a more wider area of ”biofuel processes”. I think that authors interchanged the term “biofuel” or “biodiesel” with “bioethanol”. Also introduction part should present the knowledge gaps of process control for bioethanol processes and emphasize the novelty of this work.
Response 1: The introduction was indeed too extensive towards environmental protection, especially the influence of bioethanol on the environment. In the new version this aspect has been significantly limited, and in the remaining part the production of bioethanol has been presented only in the aspect of the improvement in the energetic coefficient that is correlated to the control process optimisation.
Point 2: Is rectification column the biggest energy consumer in the bioethanol processes? What is the typical fraction of the energy consumption in a regular bioethanol plant? I think rectification column is a very common process unit in most of the chemical processes and it is not hard to control rectification column. Please justify why chose rectification column as a control application.
Response 2: In fact the rectification column is used not only in the distilling industry. The rectification process of ethanol is usually the last production stage and the energy consumption can be different. Depending on the raw material the bioethanol is made of, e.g. grains, corn, rectification constitutes less than 20% of the energetic expenses for the production [Marczak 2012]. In case of molasses, rectification takes most of energy expenses – 80%.
Point 3: The implemented controller in this work is to drive the bottom temperature to setpoint. It is much more meaningful to produce on-spec bioethanol (e.g., ethanol concentration >95%). Have the authors tried to deign the controller for a more flexible operation of rectification column.
Response 3: We believe, in order to work with more flexibility, an advanced driver based e.g. on artificial intelligence (fuzzy logic) has to be used. However, in case of columns with permanent process, the dosage of raw material depends on the concentration of ethanol, whereas in the periodically working columns setting the constant column temperature, which is strongly correlated to the distillation mixture, assures a constant concentration of bioethanol until it is exhausted in the setting. The use of such driver can be the subject of system analyses in the future.
Point 4: How did you tune control parameters (e.g., kp, Ti, Td) for each implemented controller?
Response 4: The control parameters have been adjusted using the classis Ziegler-Nichols method, because the setting optimisation process itself was not the subject of the research.
Point 5: Energy usage in the result part is the average energy usage during the whole time horizon or the energy usage at the reached steady state.
Response 5: In the research a periodically working rectification column was used, for that we can distinguish the phase of heating the setting and the steady temperature sustaining the processing. In our researches, energy consumption analysis has been carried out during the whole process, because we wanted to see how the algorithm behaves in the transition phase between these two stages.
Point 6: The transfer function representing rectification column is developed from lab-scale experimental tests. Please comment that how close/different this model is from other model in the literature.
Response 6: The developed transfer function originates from the analysis of thermal transformations, which is the initial point. The determined relationship of the transfer function that served for the work analysis of three control algorithms is typical for many thermal processes in its structure. In the related literature a description of a column can be found that has a form of differential equations for permanent processes, which is typical for the distillation industry in the large scale. There is no information about the description of a periodic rectification process.

Reviewer 3 Report
In my opinion, the manuscript processes-1075781 is a valuable scientific paper with practical and application values. The topic fully corresponds to the profile of the Processes journal. In my opinion, this is a valuable manuscript but before publication needs to be improve. My comments and suggestion below:
Abstract - The author fails to emphasize the novelty and significance of the study. An abstract summarizes, usually in one paragraph of 150-250 words or less, the major aspects of the entire paper in a prescribed sequence that includes: i) the overall purpose of the study and the research problem(s) you investigated; ii) the basic design of the study; iii) major findings or trends found as a result of your analysis; and, iv) a brief summary of your interpretations and conclusions.
Graphical abstract would be very useful for the reader. It would help in understanding the authors' intentions and the scheme of the research work carried out.
The authors should explicitly specify the novelty of their work. What progress against the most recent state-of-the-art similar studies was made in this study? Mention this in the revised manuscript sections, including abstract, introduction, and conclusions.
The introduction should show the reader more what the authors' research brings to the commonly known knowledge, which inspired them to plan and implement them, and what new they bring to science. This is completely missing and needs to be completed.
The authors did not formulate any research hypotheses. This should be the starting point for research planning. What did they expect? What were they trying to verify? Needs to be completed.
Author should also pay more attention to the practical implications of this study, outlining the challenges in the current research, future work, and recommendations. There are many problems to discuss.
Broader presentation of the shortcomings and weaknesses of the presented solutions and a stronger focus on the practical and economic aspect and the area of application is needed.
Where is the scientific discussion? Scientific discussion of the obtained results is not exist. It must be supplemented and expanded with a full comparative analysis with the currently published, current scientific works.
It is recommended to extend the comparison of the study findings under Results and Discussion with other similar published work under the results and discussion section. Currently, such a comparison is limited in this section.
The conclusions are very disappointing. The applications contain too much unnecessary information related to the purpose and methodology of the research. It needs to be corrected and changed.
Author Response
Response to Reviewer
Thank you for the valuable comments that undoubtedly emphasise the scientific character of this work and let better understand the problem we wanted to explain. We hope the corrections we made will be satisfying.
In my opinion, the manuscript processes-1075781 is a valuable scientific paper with practical and application values. The topic fully corresponds to the profile of the Processes journal. In my opinion, this is a valuable manuscript but before publication needs to be improve. My comments and suggestion below:
Point 1: Abstract - The author fails to emphasize the novelty and significance of the study. An abstract summarizes, usually in one paragraph of 150-250 words or less, the major aspects of the entire paper in a prescribed sequence that includes: i) the overall purpose of the study and the research problem(s) you investigated; ii) the basic design of the study; iii) major findings or trends found as a result of your analysis; and, iv) a brief summary of your interpretations and conclusions.
Response 1: According to the suggestion, the Abstract has been redrafted in order to emphasise the newness of the work and indicate the individual phases of the study.
Point 2: Graphical abstract would be very useful for the reader. It would help in understanding the authors' intentions and the scheme of the research work carried out.
Response 2: A graphical summary would indeed help understand the intensions of the Authors of this study. However, the discussed attempts to create such summary by the Authors introduced a lot of ambiguity in the first message. Thus, it has not been published.
Point 3: The authors should explicitly specify the novelty of their work. What progress against the most recent state-of-the-art similar studies was made in this study? Mention this in the revised manuscript sections, including abstract, introduction, and conclusions.
Response 3: The corrected paper has been supplemented with following statements:
One of the possibilities for the limitation of energy consumption in the manufacturing process of bioethanol is the optimization of the control algorithm. In the scope of the control, especially the control quality, extensive literature is available. Hereby not only classic control algorithms are considered, but also modifications of them and the artificial intelligence. The literature is also comprehensive regarding the setting optimisation of these controllers and their influence on the control signals, which is a one-criterial optimization of the control algorithms. New in this paper is the introduction of the second criterion for the assessment of the subject algorithms, corresponding to the energy consumption in energy-intensive processes. Besides, in a sense, new is the selection of the researched object, i.e. a periodically working rectification column that can be used e.g. by farmers to produce biofuel from waste. The Authors believe, researches in this area have not been carried out. Thus, it should be considered as reasonable to carry out an analysis of the influence of the applied control algorithm, among others, on the energy intensity of the process. Hereby a condition has been imposed that the methodology has to enable an analysis by modelling and computer simulation at the stage pf the technical system designing. New in the presented study is the analysis of the energy consumption reduction by selecting the optimal control algorithm. Besides, new is the execution of the analysis in question at the designing stage by modelling and computer simulation.
Point 4: The introduction should show the reader more what the authors' research brings to the commonly known knowledge, which inspired them to plan and implement them, and what new they bring to science. This is completely missing and needs to be completed.
Response 4: According to the suggestion, the introduction has been redrafted, taking into account the above-mentioned recommendations. At the same time, it has been indicated that in the available literature there are no research results for a periodic process, where the signal quality and energy consumption would be analysed regarding the control algorithm. The practical part of the research in the future will be the implementation of the control algorithm’s optimal solution in small installations to produce bioethanol in a periodically working system.
Point 5: The authors did not formulate any research hypotheses. This should be the starting point for research planning. What did they expect? What were they trying to verify? Needs to be completed.
Response 5: The corrected paper has been complemented with following contents:
The research hypothesis is: the control algorithm (the way the signal is formed by the controller) has an influence on the energy consumption in the production process of bioethanol.
Point 6: Author should also pay more attention to the practical implications of this study, outlining the challenges in the current research, future work, and recommendations. There are many problems to discuss.
Response 6: With the increase in the ecologic consciousness of the automotive market participants, there are many initiatives emerging that aim to make the transport more environment-friendly. One of the indicatives of such efforts is use of fuels obtained from renewable power sources. Ethanol from biomass belongs to this group of fuels. It can be directly applied as fuel (e-biofuel cell technology by Nissan) or as an addition to engine fuels, including the use as a component in the production process of biodiesel.
Use of ethanol as an addition to gasoline brings a lot of benefits for the natural environment. Experimental researches have proven that in influences the reduction of the emission of pollutants such as CO, SO2, CO2. It contains oxygen, so that it increases the octane number of the gasoline. It contains neither sulphur nor aromatic hydrocarbons, such as benzene, so that the exhaust gases contain less carbon monoxide, hydrocarbons, sulphur compounds and solid particles.
One of the criteria for the profitability of the production is the relationship of the energy provided in the manufacturing process the amount of energy possible to obtain while burning bioethanol.
If some specific conditions are fulfilled, the production costs of bioethanol could be reduced. Those conditions are: biofuel production from waste at the site where it is used (by interested companies with vehicle fleets or private persons, including farmers), using renewable energy, limitation of the energy intensity of the process. Thus, in this paper the possibility to reduce the energy intensity in the production process of bioethanol for biofuel by optimising the control algorithm was analysed.
Point 7: Broader presentation of the shortcomings and weaknesses of the presented solutions and a stronger focus on the practical and economic aspect and the area of application is needed.
Response 7: In this scientific report, the Authors wanted to verify the hypothesis if the control algorithm has an influence on the energy consumption in correlation with the quality of the control signal in the rectification process, which is one of the most energy-intensive processes in the production of bioethanol.
Point 8: Where is the scientific discussion? Scientific discussion of the obtained results is not exist. It must be supplemented and expanded with a full comparative analysis with the currently published, current scientific works.
Response 8: It’s a valuable comment that really emphasises the newness of the report and simultaneously it impedes the execution of a comparative analysis. The limited discussion is caused by the fact that the analysis of the control method correlated to energetic expenses for the rectification process is not sufficiently presented in the literature. The paper has been complemented with such analyses, but on the example of the thermal processes in foundry furnaces, with separate analysis of signal quality and power consumption.
Point 9: It is recommended to extend the comparison of the study findings under Results and Discussion with other similar published work under the results and discussion section. Currently, such a comparison is limited in this section.
Response 9: The recommendation has been taken into account in the attached instruction that indirectly corresponds to the rectification process, because there are no such studies, but also to the energy-intensive thermal processes such as foundry furnaces. The recommendation increases significantly the practical value of the study.
Point 10: The conclusions are very disappointing. The applications contain too much unnecessary information related to the purpose and methodology of the research. It needs to be corrected and changed.
Response 10: The indicated repetitions emphasising the purpose of the study might be indeed excessively signalled. Thus, according to the recommendation, the correction in the introducing paragraph has been carried out. However, regarding the substantial effects of the study, the Authors kept the original version, because it unambiguously summarises the effects of this work.
Round 2
Reviewer 1 Report
The manuscript has a significant improvement against the previous version.
Reviewer 2 Report
The presented version has addressed all concerns from my side. In my view, it is can be accepted in the present form.
Reviewer 3 Report
Many thanks for improving the paper. In my opinion manuscript can be publish in present form.